# ‘Selected’ Exosomes from Sera of Elderly Severe Obstructive Sleep Apnea Patients and Their Impact on Blood–Brain Barrier Function: A Preliminary Report

**DOI:** 10.3390/ijms252011058

**Published:** 2024-10-15

**Authors:** Pauline Guillot, Sebastien Celle, Nathalie Barth, Frederic Roche, Nathalie Perek

**Affiliations:** 1Gérontopôle AURA, 42000 Saint-Etienne, France; nathalie.barth@gerontopole-aura.fr; 2Inserm, U1059, Sainbiose, Faculté de Médecine Jacques Lisfranc, Université de Lyon, 42000 Saint-Etienne, France; sebastien.celle@univ-st-etienne.fr (S.C.); frederic.roche@univ-st-etienne.fr (F.R.); nathalie.perek@univ-st-etienne.fr (N.P.); 3Faculté de Médecine Jacques Lisfranc, Université Jean Monnet, 42000 Saint-Etienne, France; 4Physiologie Clinique et de l’Exercice, Centre Visas, CHU Saint Etienne, 42000 Saint-Etienne, France; 5Chaire Santé des Ainés, Ingénierie de la Prévention, Université Jen Monnet, 42000 Saint-Etienne, France

**Keywords:** elderly, obstructive sleep apnea, exosomes, biomarkers, blood–brain barrier, tau, amyloid, tight junction proteins

## Abstract

Obstructive sleep apnea syndrome (OSAS) affects a large part of the aging population. It is characterized by chronic intermittent hypoxia and associated with neurocognitive dysfunction. One hypothesis is that the blood–brain barrier (BBB) functions could be altered by exosomes. Exosomes are nanovesicles found in biological fluids. Through the study of exosomes and their content in tau and amyloid beta (Aβ), the aim of this study was to show how exosomes could be used as biomarkers of OSAS and of their cognitive disorders. Two groups of 15 volunteers from the PROOF cohort were selected: severe apnea (AHI > 30) and control (AHI < 5). After exosome isolation from blood serum, we characterized and quantified them (CD81, CD9, CD63) by western blot and ELISAs and put them 5 h in contact with an in vitro BBB model. The apparent permeability of the BBB was measured using sodium-fluorescein and TEER. Cell ELISAs were performed on tight junctions (ZO-1, claudin-5, occludin). The amount of tau and Aβ proteins found in the exosomes was quantified using ELISAs. Compared to controls, OSAS patients had a greater quantity of exosomes, tau, and Aβ proteins in their blood sera, which induced an increase in BBB permeability in the model and was reflected by a loss of tight junction’ expression. Elderly patients suffering severe OSAS released more exosomes in serum from the brain compartment than controls. Such exosomes increased BBB permeability. The impact of such alterations on the risk of developing cognitive dysfunction and/or neurodegenerative diseases is questioned.

## 1. Introduction

Obstructive sleep apnea syndrome (OSAS) is characterized by repeated partial or complete upper airway obstruction during sleep. These obstructions alter cyclical alveolar ventilation, leading to episodic hypoxemia and hypercapnia, as well as sleep fragmentation. OSAS is a very common syndrome among the elderly, with an increase in prevalence from the age of 60 years, but it remains a syndrome that is not very often diagnosed. Between 37.5–62% of people over 60 years may suffer from this syndrome [1]. OSAS is associated with an increased risk of diseases, including cardiovascular, cerebrovascular, and cognitive impairment [2,3], potentially leading to memory loss and loss of autonomy, particularly in the vulnerable elderly [4,5,6]. Indeed, recent research shows that vitamin D deficiency can worsen sleep disorders, particularly in hypertensive patients with OSA, by influencing sleep quality and cognitive function [7]. At the heart of consideration is the definition of OSAS, commonly defined according to an apnea-hypopnea-index (AHI) > 5. AHI is currently used as a reference measurement tool, with reference values enabling us to distinguish between non-apneic subjects with an AHI < 5, subjects with mild apnea whose AHI is between 5–15, subjects with moderate apnea whose AHI is between 15–30, and finally severe apneics whose AHI > 30 [3]. Also, OSAS is defined according to the oxygen-desaturation-index (ODI), which is the number of episodes of oxygen desaturation per hour of sleep during which blood oxygen has fallen by ≥3% [8]. According to the American Academy of Sleep Medicine (AASM), the diagnosis of OSAS is based on AHI. An AHI ≥ 5 with symptoms indicates mild OSAS, while an AHI ≥ 15 is diagnostic of moderate to severe OSAS, even in the absence of symptoms or comorbidities [9]. European recommendations for the diagnosis of OSAS are also based on AHI. AHI ≥ 5 is diagnostic of mild OSAS, but diagnosis may be more focused on the presence of clinically relevant symptoms. An AHI ≥ 15 allows the diagnosis of moderate to severe OSAS, with greater consideration for associated comorbidities and impact on quality of life compared with the AASM recommendations [10].

On the basis of a cutoff at 65–95 years, prevalence rates are 62% for an AHI > 10, 44% for an AHI > 20, and 24% for an AHI > 40 [3,11]. In the elderly population, this OSAS pathology is still rarely diagnosed, and its consequences are underestimated, such as cognitive decline. The reference treatment is continuous positive airway pressure (CPAP). One of the main obstacles to treatment with CPAP is patient compliance, as this is a long-term, restrictive treatment that patients often stop. When patients do not use CPAP enough, clinical effects are compromised. Furthermore, the efficacy of CPAP in treating certain neurological disorders remains to be demonstrated, although there is evidence to suggest that CPAP interventions may improve cognitive function [12]. Identification of apneic patients with a high risk of cognitive decline remains difficult and sometimes after a long time. Thus, an improved understanding of the cellular and molecular mechanisms that lead to central nervous system (CNS) dysfunctions is needed in order to develop therapeutic strategies. The hypothesis is that a disruption of the blood–brain barrier (BBB) has emerged and may contribute to cognitive impairments in OSAS associated with intermittent hypoxia (IH) and sleep fragmentation [13,14]. Circadian rhythm appears to be closely linked to BBB permeability, and a poor cycle could promote the passage of toxic substances, increasing the risk of neurodegenerative diseases and neuroinflammation [15]. In addition, a pilot study on proteins involved in circadian rhythm regulation revealed that the latter may also play a role in IH-related alterations in glucose metabolism in OSAS patients [16].

Indeed, OSAS influences the permeability of the BBB and could promote the passage of toxic substances to the brain, increasing the risk of neurodegenerative diseases and neuroinflammation [15]. This increase in permeability is linked to increased inflammatory signaling and a decrease in tight junction proteins [17,18].

The BBB is a multicellular vascular structure that acts as a diffusion barrier to prevent the entry of most compounds forming blood into the brain, thereby allowing the maintenance of brain homeostasis. Endothelial cells form the walls of the brain capillaries and represent the anatomical basis of the BBB [19]. The BBB is mainly composed of endothelial cells linked together by tight junctions (TJs). It is also composed of astrocytes, which play a major role in the maintenance of the junctions, as well as pericytes and neurons [20].

The mechanism connecting sleep disturbances and memory seems to be the nightly elimination of toxins or misfolded proteins such as amyloid beta peptides (Aβ) and tubulin-associated unit protein (tau) accumulated in the brain during the previous day. IH affects the production and clearance of Aβ and tau proteins, linked to neurodegenerative diseases [21,22]. IH increases cytokine production, stimulating beta- and gamma-secretase enzymes, and is responsible for amyloid precursor protein cleavage and Aβ accumulation [23]. Furthermore, SF disrupts the cycles required for Aβ and tau clearance and alters BBB tight junctions, particularly via Aβ peptides [24,25].

More recently, it has been proposed that exosomal cargo containing Aβ and tau provides cues to mechanistic pathways while also serving as biomarkers of neurocognitive risks in patients [26]. It has been demonstrated that Alzheimer’s disease (AD) and sleep apnea share common features like inducing tau and Aβ protein accumulation [11,27,28,29].

Exosomes are a class of endosome-derived membrane vesicles shed by cells, which contain proteins and other constituents of their cellular origin. They are known for their potential involvement in neurodegenerative diseases via their passage through the BBB [30,31,32,33]. Their main role is inter-cellular communication. According to their origin, their composition differs [34]. Exosomes are found in high quantities in the bloodstream of apneic patients [35]. They represent a valuable source of central nervous system-derived biomarkers that can be isolated from blood [36]. The Aβ and tau proteins are usually assayed in the CSF. However, the cost of repeating CSF sampling and the invasive nature of this method underlines the importance of developing cheaper and less invasive tests to predict cognitive risks, such as blood tests [27]. Aβ and tau proteins are very difficult to detect in soluble form in the blood, and exosomes could be a good method of assaying these proteins [37].

Based on the leading hypothesis that OSAS produces altering exosomal cargo in serum, and that these altering exosomes increase BBB permeability and promote pathophysiological mechanisms that have been implicated in cognitive deficits, we proposed to explore this major hypothesis.

Therefore, in this study, we were interested in exosomes carrying the tau and Aβ proteins, both from the cerebral compartment. These two proteins have also been shown to be involved in neurodegenerative diseases such as AD [12] and provide cues to mechanistic pathways while also serving as biomarkers of neurocognitive risks in patients. It has also been proposed that exosomal cargo may serve as biomarkers of neurocognitive risks in patients suffering from OSAS. If exosomes cargo contain tau and Aβ proteins, the detection of these exosome signature proteins supports the possibility of their role in the generation of cognitive decline.

Our project was to establish serum biomarkers and prevention methods to determine the phenotype at risk of OSAS by establishing a score of risk with transdisciplinary approaches. The objective of this study was to show how exosomes could be used as biomarkers of OSAS and neurocognitive disorders following OSAS by showing their presence in high quantity in the blood sera of elderly patients with apnea, as well as their origin from the brain compartment by the presence of tau and Aβ proteins [26,38].

In addition, this study evaluated the effect of exosomes from a group of elderly apneic patients versus control patients on an in vitro model of the BBB using permeability measurements. According to the results of this study, we could consider use of exosomes in the diagnosis of OSAS in neurocognitive diseases, with exosomes being secreted well before the onset of symptoms. This is in line with the idea of preventing cognitive disorders and loss of autonomy in the elderly and to promote aging well.

## 2. Results

### 2.1. PROOF Subjects Characteristics

Our selected population comprised of elderly patients (Table 1). Two groups: OSAS with severe apnea (AHI > 30) and non-apneic subjects considered as control (AHI < 5). These two groups were similar in terms of age but different in terms of gender, AHI, Oxygen desaturation index (ODI), which is the number of times per hour of sleep that the blood oxygen level drops by a certain degree from baseline, and their hypoxemic load. Each of the participants underwent polygraphy, which allowed the calculation of the AHI index: the number of apneas/hypopneas made per hour of sleep, describing the severity of the apnea. Blood samples were then collected in the same time frame for all participants.

### 2.2. Characterization of Exosomes—Western Blot and Homemade ELISA

Following precipitation, the exosomes were further purified using a chromatography column. Analysis with the Zeta Sizer confirmed the purification, revealing the presence of exosomes with a size of 108.7 nm and a concentration of 2.3 × 10^7^ particles/mL (Figure 1).

Two methods were used to characterize and then perform comparative quantification on exosome samples obtained by precipitation and chromatography for each patient. Our results showed a higher amount of tetraspanins (CD81, CD9, and CD63) characteristic of exosomes (Figure 2 and Figure 3); ELISA’s of CD81 (** *p* = 0.0026 < 0.01) in OSAS patients (AHI > 30) vs. controls. Similarly, a higher amount of Aβ protein was observed in OSAS patients (** *p* = 0.0014 < 0.01) vs. controls. On the other hand, there was no significant difference in tau protein abundance in OSAS vs. controls (*p* = 0.1453 > 0.01), although there was a slight increase in tau in OSAS patients (Figure 3). Five points stand out (Figure 3 red points), reflecting a high level of CD81, Aβ, and tau proteins in patients with OSAS and patients with an AHI ranging from 30.3–63.6 (AHI mean: 48.46).

### 2.3. Impact of Exosomes on the BBB Model—Monolayer of HBEC-5i Endothelial Cells in Conditioned Medium of HA Astrocytes

The protocol was optimized according to the literature and cytotoxicity tests. A time of 5 h and a concentration of 10 μg enabled us to observe the effects of exosomes in the model while maintaining optimal culture conditions. This condition was applied to all permeability tests.

#### Permeability Tests—Na-F and TEER

After 5 h of incubation with 10 μg exosomes on the luminal side of the model, permeability tests were carried out. Firstly, using Na-F, we observed a significant increase in permeability in OSAS (AHI > 30) vs. controls (** *p* = 0.0049 < 0.01) (Figure 4a). The same observation was made when permeability was measured using the TEER method (*** *p* = 0.0009 < 0.01) (Figure 4b). Five points stand out (Figure 4 red points), reflecting a sharp increase in permeability in OSAS patients, being patients with an AHI between 30.3 and 63.6 (AHI mean: 48.46).

### 2.4. TJs Expressions: ZO-1, Claudin-5 and Occludin

The expression of junctions (ZO-1, claudin-5, and occludin) was studied after 5 h incubation of HBEC-5i cells cultured in 96-well plates with 10 μg of exosomes. Claudin-5 expression decreased significantly (*** *p* = 0.0008 < 0.01) in OSAS patients (AHI > 30) compared to controls (Figure 5b). ZO-1 expression (** *p* = 0.001 < 0.01) decreased less than claudin-5 expression but significantly in OSAS patients (AHI > 30) compared to controls (Figure 5a). Concerning occludin expression, despite a slight decrease in expression in OSAS patients (AHI > 30) (Figure 5c), there were no significant differences compared to controls (*p* = 0.3126 > 0.01).

### 2.5. Correlation Tests

The following data were correlated using Pearson’s coefficient: AHI, ODI, CD81 expression, tau expression, Aβ expression, apparent permeability: NaF, ZO-1 expression, claudin-5 expression, and occluding expression. We performed correlation significance tests and discussed Pearson correlation coefficients with a *p*-value < 0.01 (Figure 6a).

There was a strong positive correlation between AHI and ODI (R^2^ = 0.92), reflecting the fact that the higher the AHI, the higher the ODI and, therefore, the greater the oxygen saturation. AHI was also positively correlated with CD81 expression (R^2^ = 0.71), showing that the more severe the patient’s apnea, and therefore the higher the AHI, the greater the number of exosomes found in the patient’s blood serum. Since AHI was also positively correlated with permeability (Na-F) (R^2^ = 0.66) (Figure 6b), we could understand that a high AHI induced a high secretion of exosomes as well as an alteration of the BBB, also seen through the negative correlation of junctions: claudin-5 (R^2^ = − 0.65). ODI is, therefore, positively correlated with CD81 expression (R^2^ = 0.67) and with permeability (Na-F) (R^2^ = 0.51). ODI is also correlated with Aβ expression (R^2^ = 0.51), which is a fairly low coefficient but with a *p*-value < 0.01. As AHI, ODI is negatively correlated with claudin-5 (R^2^ = − 0.65).

CD81 expression was also strongly and positively correlated (R^2^ = 0.89) with permeability (Na-F) (Figure 6c), allowing us to assert that a high AHI induced greater CD81 expression, which in turn is at the origin of BBB alteration.

Finally, we note that the expression of Aβ correlated with a decrease in the expression of the junctional protein ZO-1 (R^2^ = − 0.69). No correlation was observed for tau protein expression or for occludin junction protein expression.

## 3. Discussion

Our study aimed to investigate the link between circulating exosomes and their cargos in brain endothelial dysfunction in severe OSAS elderlies. The integrity of the BBB is crucial for the maintenance of efficient cerebral homeostasis. Disruption of this integrity has already been shown in aging and is associated with various neurological disorders as well as OSAS [39,40,41].

Indeed, recent studies have demonstrated that specific diseases or intermittent chronic stress, such as sleep apnea/hypopnea, can disrupt brain endothelial TJs [41] and affect cognition [14]. It has been shown in our laboratory that blood serum from elderly apneic patients in contact with an in vitro model of the BBB induced an alteration of the BBB, resulting in an increase in permeability of the model and opening the in vitro BBB barrier [41]. Previous work suggested that OSAS may play a key role in the emergence of cognitive disorders, notably through increased BBB permeability via modulation of Nrf2 expression and dysregulation of ABC transporters [42].

Our study highlighted the potential of serum exosomes as biomarkers of OSAS, revealing significantly higher levels of CD81, CD9, and CD63 proteins in elderly OSAS patients compared to elderly healthy subjects. Elderly patients with an AHI > 30 and a mean of 48.5 had much higher quantities of exosomes in their blood serum, making our study original since we understand that there is a potential correlation threshold between AHI, ODI (i.e., the severity of the sleep-related breathing disorders) and the number of exosomes crossing the BBB and found in blood serum. In fact, AHI and ODI have a Pearson coefficient of 0.909, allowing us to correlate the number of exosomes with ODI as a marker of intermittent hypoxemic load.

The study of exosomes and their cargos could provide significant valuable insight into the mechanism behind cell-to-cell communication and disease development and progression in OSAS. Exosomes are extracellular vesicles involved in intercellular communication. Their implications could be crucial in the pathogenesis of neurological disorders by facilitating a pro-inflammatory phenotype and compromising the integrity of the BBB [26]. Our in vitro experimentation revealed that exposure of BBB models to similar amounts of exosomes from apneic vs. control elderly subjects significantly increased BBB permeability, suggesting a specific OSAS-related detrimental interaction of exosomes on the barrier. The present study showed that exosomes from elderly patients with severe OSAS induced significant increases in permeability of the BBB model, reflecting the high BBB toxicity of exosomes from patients with such diseases.

Processes of BBB alteration could be mediated via exosomal-related biological activities that directly impact the BBB permeability and functionality [25,30]. The opening of the BBB and subsequent infiltration of serum components to the deep brain can lead to a host of processes resulting in progressive synaptic, dendritic neural soma, dysfunction, and detrimental neuroinflammation environmental changes as microglial pre-activation previously demonstrated in a mouse OSAS model [43]. Such processes have been implicated in different diseases, including vascular dementia, AD, stroke, Parkinson’s disease, multiple sclerosis, amyotrophic lateral sclerosis, severe hypoxia, ischemia, and diabetes mellitus [44].

The BBB is an essential structure protecting the CNS by regulating blood flow to the brain, and paracellular transport is notably regulated by junction proteins such as ZO-1, claudin-5, and occludin. A decrease in expression of these junctions in contact with exosomes, in subjects with an AHI > 30, leads to BBB embrittlement [45,46].

ZO-1 and claudin-5 proteins play a major role in BBB development [47]. ZO-1 and claudin-5 show a significant decrease in expression when endothelial cells encounter (direct contact exposure) exosomes in elderly apneic patients, compared to controls, reflecting BBB dysfunction. Alteration of ZO-1 expression has already been shown to be associated with Aβ protein expression in AD [48]. On the other hand, there was no significant difference concerning occludin expression, as is often the case in the literature [49], despite the finding of an alteration in these TJs involved in maintaining BBB integrity.

Studies have established possible correlations between OSAS and neurodegenerative diseases such as AD [12], notably by investigating levels of the proteins Aβ and tau, marked by their accumulation in the brains of patients with AD [50,51]. These proteins, usually measured in cerebrospinal fluid, can also be detected in blood via exosome analysis, enabling the use of a less invasive and equally significant method [32,52]. Indeed, as well as being less invasive for the patient, increasingly research is looking into the use of blood biomarkers, notably in AD and other neurodegenerative diseases, to enable better and often earlier diagnosis [53]. Fiandaca et al. [27] support the fact that detection in individuals of elevated amyloid and tau exosomal proteins in the blood could reflect a current or future neurodegenerative disease.

Markedly elevated levels of Aβ protein in exosomes from elderly apneic subjects suggest a possible direct correlation between OSAS and neurological changes, corroborating the findings of Sun et al. [12]. These authors demonstrated that intermittent hypoxia causes a decrease in alpha-secretase activity and an increase in beta-secretase, thereby increasing the production of Aβ in the brain and its secretion into exosomes as they form in the brain. Although we observed an increase in tau protein in the exosomes of apneic subjects, it did not reach statistical significance in comparison with healthy non-apneic subjects, possibly due to the high prevalence of increased plasmatic level this protein in our elderly cohort, including in the control group; and indeed this hypothesis was proposed in the literature [12,54].

OSAS induces sleep fragmentation, which disrupts sleep and the recycling of metabolic waste products. Reduced glymphatic activity hampers the process of clearing Aβ and tau proteins, enabling us to make a direct link between OSAS and AD [55]. In addition, intermittent hypoxia causes oxidative stress and inflammation, potentially contributing to the accumulation of these neurotoxic proteins [56,57]. As before, exosomes from elderly patients with an AHI > 30 and a mean of 48.46 induced significant increases in Aβ within exosomes. This suggests that there is an AHI or hypoxemic load threshold above which the patient is at a high risk of developing cognitive impairment and that exosomes alter the BBB in a way that is clearly more toxic in elderlies or adults above this threshold. In fact, the more exosome concentration the patients have, the more likely they are to alter the BBB, which will no longer play its filtration role, and the greater the quantities of Aβ found inside the exosomes. Moreover, the high levels of Aβ are also associated with poor recycling during sleep, as mentioned above. With regard to tau protein, elderly patients with severe OSAS showed a marked but non-significant increase in the presence of such biomarkers.

This study highlights the possible role of exosomes in mediating negative OSAS impacts on the BBB. This has significant implications for understanding how the risk of developing cognitive disorders might be assessed and anticipated. The use of exosomal biomarkers from blood serum in combination with AHI is particularly important for detecting high-risk individuals. These findings suggest that exosomes could be used for early neurological risk assessment, with the ability to detect changes up to 10 years before the emergence of clinical symptoms [27].

It is essential to show the importance of early detection of markers of BBB dysfunction, such as exosomes carrying Aβ and tau proteins, for the introduction of therapeutic strategies aimed to limit or stop progression to cognitive impairment. Early treatment of OSAS, notably with CPAP, has demonstrated a reduction in Aβ and tau levels, highlighting the reversible potential of these biomarkers in response to targeted therapeutic intervention [12].

However, our study is limited by the fact that our target population is the elderly. It might be relevant to compare our results with those of a cohort of younger OSAS patients, as we hypothesize that the amount of tau protein recovered is age-related, so we might be able to observe differences in younger subjects. Furthermore, a study of the amount of phosphorylated tau could have provided us with additional information, but this protein is more difficult to assay. Elderly subjects are exposed to more comorbidities and, therefore, naturally have a greater quantity of serum exosomes. The gender factor also comes into play, with a higher proportion of women than men. It would also be useful to supplement our study with an analysis of synchronized neuropsychological tests and brain MRI or PET imaging [58]. It should be noted that our tests were carried out on in vitro models of the BBB made up of an hBEC-5i cell type and conditioned astrocyte medium; however, to get as close as possible to reality, we would like to develop a tri-culture model involving pericytes which are known to play an important role in aging, which is our target population [59]. According to the literature, we have chosen to expose in vitro BBB models to 10 μg of exosome concentration, which does not necessarily reflect the amount each individual possesses. Exosomes contain a wide variety of molecules, including miRNAs (e.g., miR-132), and the study of these miRNAs could enable us to make even more precise studies with a view to developing new diagnostics and treatments.

## 4. Materials and Methods

### 4.1. Subjects

The blood sera used was from the PROOF cohort study in Saint-Etienne (France). This cohort was set up in 2001, and 1011 subjects aged 65 years at the time of inclusion were recruited. The aim of this study was to evaluate the role and the alteration of the autonomic nervous system in the occurrence of cardiovascular or cerebrovascular events and of dementia over 20 years of follow-up [60]. The PROOF study was made possible by an association with the SYNAPSE study, which focused on approximately the same sample group on sleep-disordered breathing and CNS incident disorders. Thirty sera were selected for our study: 15 from elderly patients with severe apnea OSAS (AHI > 30) and 15 from elderly patients without apnea and no OSAS (AHI < 5), according to the European Respiratory Society (ERS). We designed a comparative study between two groups of subjects based on the PROOF cohort, in which we had sufficient biological data for 15 patients with severe apnea. To maintain parity between the groups, we randomly selected 15 control subjects from a larger sample.

For this first phase of the study, we deliberately chose to compare the two extremes of AHI in order to observe marked differences between the groups. This will facilitate the identification of possible effects of severe apnea.

Several studies have already been performed using the PROOF cohort, approved by the CCPRB Rhône Alpes Loire IRB N532016 and IRB2002/22/CHUSETE, and agreement was given by the CNIL (National Committee for Information and Liberty) for data collection. The participants gave their written consent to participate in the PROOF studies.

Subjects in this study were selected on the basis of polygraphy findings, taking into account the AHI. We also have biological data such as total cholesterol, LDL, blood glucose and CRP levels. It is important to note that we have no information on whether participants were taking cholesterol- or glucose-lowering drugs. However, assays for these biomarkers were similar between the apneic and control groups, indicating the absence of confounding factors at the time of sampling (Table 1).

There was a noticeable imbalance between the number of men and women in the study, as the women were postmenopausal; however, this bias was partly neutralized by the age of the participants. Inclusion criteria were an AHI < 5 for the control group and >30 for the apneic group, with all subjects in good health with no signs of inflammation (confirmed by cholesterol, blood glucose, and CRP levels). Patients with neurodegenerative diseases or other comorbidities were excluded.

### 4.2. Biological Samples

Blood samples were collected in the morning after being recorded using polygraphy using a polygraphic system (HypnoPTT, Tyco Health-care, Puritan Bennett, Pleasanton, CA, USA) [61]. Then, after centrifugation and recovery of the serum, it was immediately frozen at −80 °C after having made each sample anonymous according to good practices. Clinical trial numbers of SYNAPSE and PROOF studies are NCT00766584 and NCT 00759304.

### 4.3. Chemicals and Reagents

#### 4.3.1. Cell Culture

Endothelial cell growth supplement (ECGS) was obtained from Sigma-Aldrich; fetal bovine serum (FBS) was from Eurobio Scientific; trypsin, amphotericin B, and penicillin-streptomycin solutions were from Corning; human brain endothelial cells HBEC-5i were obtained from ATCC; human astrocyte cells HA and astrocyte medium AM were from CliniSciences; GIBCO Dulbecco’s Modified Eagle Medium/Nutriment Mixture (DMEM-F12) was from Fisher Scientific; 24-well inserts (transparent PET membrane, 0.45-μm pore diameter size) and all consumable for cell culture came from D.Dutscher.

#### 4.3.2. Permeability Measurement

EVOM volt ohmmeter system was purchased from Sigma-Aldrich (Burlington, MA, USA); all compounds for Ringer HEPES buffer and sodium-fluorescein sodium (Na-F) were from Sigma-Aldrich.

#### 4.3.3. Exosome Isolation & Characterization

EX02-50-Exospin blood with 48 columns was from Cell Guidance. The bicinchoninic acid (BCA) protein assay kit for low concentrations was from Abcam. MiniPROTEN Tetracell was from Biorad, Pierce ECL Western Blotting Substrate, PageRuler Plus Prestained Lader, Phosphatase and proteases inhibitors, and 0.2-μm PVDF membrane were from Thermo Fisher Scientific (Waltham, MA, USA), RIPA buffer, blocking buffer (with BSA and PBS Tween) and all compounds for Laemeli solution were from Sigma-Aldrich.

#### 4.3.4. ELISA

Recombinant human tau441 protein from Abcam, the Aβ and CD81 proteins, came from the following kits: the anti-β-Amyloid kit was from Thermo Fisher Scientific, and the ExoELISA-ULTRA CD81 kit was from Systeme Biosciences. Tetramethylbenzidine (TMB) was from Sigma-Aldrich. ELISA buffer kit was purchased from Thermo Fisher Scientific.

For all antibodies, see Table 2.

### 4.4. Blood Serum Exosome Isolation and Purification

Exosomes were isolated from human sera according to the manufacturer’s protocol by Cell Guidance Systems (EX02-50). The end step was purification by passage to the steric size exclusion chromatography column. The column is composed of pores about 30 nm in resin according to the kit recommendations. The exosomes could be used immediately; for long term storage the exosomes were frozen at −80 °C.

### 4.5. Protein Measurements

Isolated exosome samples were dispersed in 96-well plates, to which a BCA solution was added. The realization of a standard range from a solution of BSA at 2 mg/mL provided in the kit, to which a solution of BCA was also added in an equivalent quantity to what was added to the samples of exosomes. This will make it possible to establish a standard linear curve in order to calculate the total protein concentration of each sample. After incubation for 2 h at 37 °C, a reading was carried out at a wavelength of 560 nm.

### 4.6. Exosomes Characterization

#### 4.6.1. Western Blot

The characterization of exosomes was performed using the western blot method, a semi-quantitative technique allowing the detection of specific proteins within a sample. 30 μg of exosomes were used per migration well. The freshly isolated exosome samples were sonicated and mixed with 1X RIPA lysis buffer and 2X Laemmli solution before being introduced into 1.5-mm SDS-PAGE gel wells (15% separation gel and 4% concentration gel). Migration was performed for 30 min at 0.04 A and 1 h at 0.07 A. The next transfer was performed on a 0.2-μm PVDF membrane for 1 h at 100 V. Subsequently; the membrane was saturated for 1 h in a blocking buffer made from BSA powder and PBS Tween20. The PVDF membrane was then incubated overnight at 4 °C with antibodies to CD81, CD63, CD9, tau, and GAPDH at 1/100 dilution and to Aβ at 1/200 dilution. After washing, the PVDF membrane was incubated for 1 h at room temperature with the secondary antibody m-IgGκBP-HRP at 1/500 dilution.

Revelation was performed by chemiluminescence using an ECL kit.

#### 4.6.2. Homemade ELISA

The western blot characterization method was completed with an ELISA method via the implementation of an inhouse anti-CD81, Aβ and tau ELISAs (results compared with the anti-CD81ExoELISA-ULTRA CD81 kit (System Biosciences: EXEL-ULTRA-CD81-1) and the anti-β-Amyloid 42 kit (Thermo Fisher scientific: KHB3441).

A standard range was performed using CD81, Aβ, and tau standard proteins, giving us standard curves. In 96-well plates, the wells were coated with a mixture of 50 μg of exosomes and phosphate coating buffer for 2 nights at 37 °C. Exosomes were incubated for 1 h with antibody anti-CD81 at 1/500 dilution and antibodies to Aβ and tau at 1/1000 dilution. Then, the HRP secondary antibody diluted to 1/10,000 was added to each well and incubated for 1 h at room temperature. Revelation was performed with TMB 10 min. Then, a stop solution was added to stop the reaction. The plate was read with an absorption spectrophotometer at a wavelength of 450 nm. Coating buffer, wash buffer, assay buffer (for antibody dilution), TMB, and stop solution were supplied in the ELISA buffer kit (Thermo Fisher Scientific).

#### 4.6.3. Nanoparticle Tracking Analysis (NTA)

A few millimeters of exosomes diluted in PBS were analyzed using a Zeta Sizer instrument (ZetaView, Particle Metrix Twin Colocalization, Excilone, Elancourt, Île-de-France, France). The exosomes were derived from the blood serum of an elderly apneic patient. During the analysis, the temperature was 29.99 °C, pH 7.0, and conductivity was 15,000 μS/cm. The instrument settings included a laser wavelength of 488 nm, a diffusion filter, a sensitivity of sensitivity 80.0, and a shutter speed of 100.

### 4.7. In Vitro BBB Model

The BBB model was composed of endothelial cells HBEC-5i cultivated in PET transwells with 0.45-μm pores in contact with HA-conditioned medium as previously described [19,41]. Initially, HBEC-5i were cultivated in DMEM/F12 HAM, containing 1% antibiotic solutions (amphotericin B and penicillin-streptomycin), 10% FBS, and 40 μg/mL of ECGS. HBEC-5i must be used before passage number 20 to maintain all BBB properties necessary to our model. Regarding HA, cells were initially cultured in AM supplemented with 1% penicillin-streptomycin solution, 1% AGS, and 2% FBS. Hereafter, HA was cultivated with the same medium as HBEC-5i. HA and HBEC-5i were incubated (5% CO_2_ and at 37 °C). The BBB model was composed of 2 × 10^5^ HBEC-5i per insert, in contact with a conditioned medium for 14 days: the time necessary to see an optimal permeability (*P_app_*) of the model, which was stable for 5 days. The HA-conditioned medium corresponded to the medium that had remained in contact with HA for 48 h. We noted that the medium was renewed every 2 days, and this was interesting because of all factors and chemical compounds released by HA, which may have effects on endothelial cells and on our BBB model permeability [19].

### 4.8. Permeability Measurements—Barrier Properties

#### 4.8.1. Trans Endothelial Electrical Resistance (TEER)

To analyze the effects of exosomes on the BBB, first, we tested the tightness of our in vitro BBB model under two conditions: inserting it in contact with exosomes from the blood serum of elderly apneic vs. elderly control. 10 μg of exosomes were added to the luminal upper side endothelial cells. The exosomes were mixed with DMEM-F12 medium without FBS and incubated for 5 h. Then, we tested the TEER using an EVOM resistance meter on the endothelial cell monolayer. This method consists of calculating the potential difference between the two poles using electrodes: one electrode on the luminal side and a second on the abluminal side. For each well measurement, three values were noted. After measurement, the following formula was applied:(TEER − control TEER) × 0.33

With 0.33 being the membrane area and the control TEER as a blank filter without cells.

Ideally, the TEER should be around 40 Ω/cm^2^, as previously described [19].

#### 4.8.2. Sodium-Fluorescein

After measuring the TEER, we performed a new method to test the apparent permeability *P_app_* of our model, using Na-F, which is a hydrophilic fluorescent molecule (MW: 376 Da). A solution of Ringer HEPES buffer diluted with 10 μg/mL of Na-F were placed onto the luminal upper side for 1 h and incubated, while the abluminal side contained only Ringer HEPES solution.

Then the solution was removed from the abluminal side, and the fluorescence was measured using a fluorescence spectrophotometer (Fluoroskan AscentTM, Thermo Fisher Scientific), with 485-nm excitation and 530-nm emission wavelengths. *P_app_* is measured in cm/s and is calculated using the formula used in our previous laboratory study [41].
$$P_{app}=\frac{V_{r}}{C0}.\frac{1}{S}.\frac{C1}{t}$$

*P_app_* is the apparent permeability, *V_r_* is the volume of medium in the abluminal side, *S* is the monolayer’s area, and *C*0 and *C*1 are the concentration of the fluorescent compound in the luminal chamber at *t* and in the abluminal side after *t* time -h of incubation, respectively.

### 4.9. Protein Expression and Quantification

Endothelial cells were seeded at a density of 2 × 10^4^ HBEC-5i per well into 96-well permeable plates and incubated with DMEM-F12. At confluence, 10 μg of exosomes were added to each well for 5 h. Then, cells were fixed with 4% paraformaldehyde and permeabilized with a solution of H_2_O_2_ and methanol. ZO-1, claudin-5, and occludin expressions were analyzed with antibodies of diluted ZO-1 (4 μg/mL), claudin-5 (2 μg/mL), and occludin (1 μg/mL), respectively, at room temperature for 2 h. Secondary IgG antibody diluted 1/2500 for anti-rabbit, and 1/500 for anti-mouse was incubated at room temperature for 2 h. Then, TMB substrate was added to each well for 15 min. To finish, hydrochloric acid was added, and the reaction was stopped before measurements were taken at 450 nm.

### 4.10. Statistics Analysis

Statistical analysis was realized using an unpaired *t*-test by GraphPad Prism 9.5 software between AHI < 5 and AHI > 30 for each of the ELISAs, permeability measurements, and cell ELISAs. The differences between means were considered to be significant when *p* values were <0.05. Results are presented as sample mean +/− SEM.

Statistical analysis was also performed using Pearson correlation coefficient R^2^ tests between parameters: AHI, ODI, CD81 tetraspanin, tau protein, Aβ protein, Na-F permeability, ZO-1 protein, claudin-5 protein, and occludin protein. Only Pearson coefficients with a *p*-value < 0.01 are displayed and considered significant.

## 5. Conclusions

Our results establish a probable link between severe OSAS and BBB dysfunction in the elderly. Exosomes could represent an interesting biomarker of the dysfunction of BBB in such patients, opening the way to innovative diagnostic and therapeutic approaches based on serum exosome analysis. Further research is needed to elucidate the precise mechanisms by which exosomes alter BBB permeability and to develop clinical methods for the routine isolation and characterization of these vesicles in order to better personalize the management of OSAS patients. An initial idea for thresholding severe apneic subjects could help predict the alteration of the BBB and possibly the consecutive cognitive dysfunctions.

## Figures and Tables

**Figure 1 ijms-25-11058-f001:**
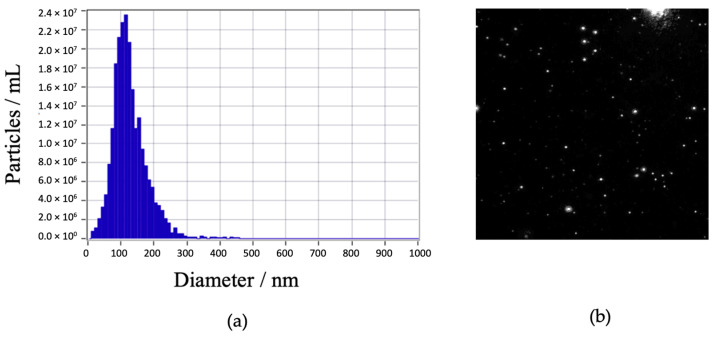
Characterization of exosomes using NTA zeta sizer instrument. (**a**) Particles in mL are a function of diameter in nm, (**b**) Visualization of exosomes.

**Figure 2 ijms-25-11058-f002:**
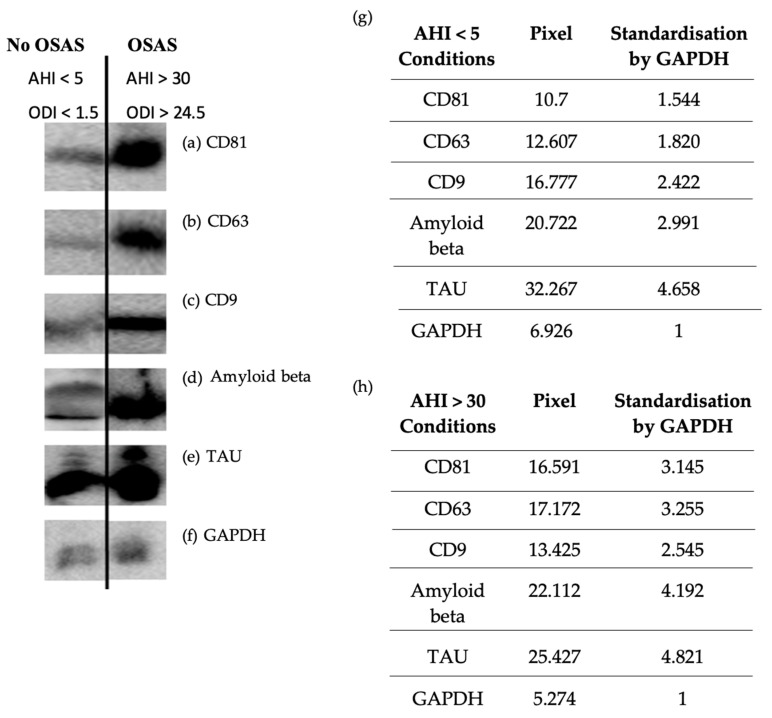
Western blot analysis of the marker protein CD81 (**a**), CD63 (**b**), CD9 (**c**), Amyloid beta (**d**), Tau (**e**), and GAPDH as control (**f**) on exosomes isolated from patients without OSAS (AHI < 5) (**g**) vs. patients with OSAS (AHI > 30) (**h**). Semi-quantification normalized with GAPDH (**g**,**h**), results obtained and analyzed with ImageJ.

**Figure 3 ijms-25-11058-f003:**
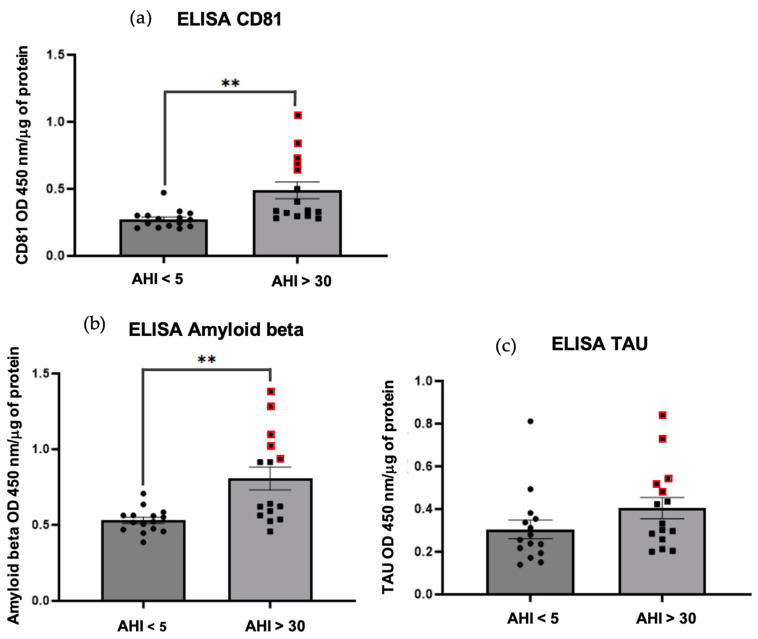
Homemade ELISAs. Results are represented as mean value +/− s.e.m (n = 15 triplicate). Expression of CD81 protein (**a**) ** *p* = 0.0026 < 0.01 between AHI < 5 vs. AHI > 30. Expression of Amyloid beta protein (**b**) ** *p* = 0.0014 < 0.01 between AHI < 5 vs. AHI > 30. Expression of total TAU protein (**c**) *p* = 0.1453 > 0.01 is not significant between AHI < 5 vs. AHI > 30.

**Figure 4 ijms-25-11058-f004:**
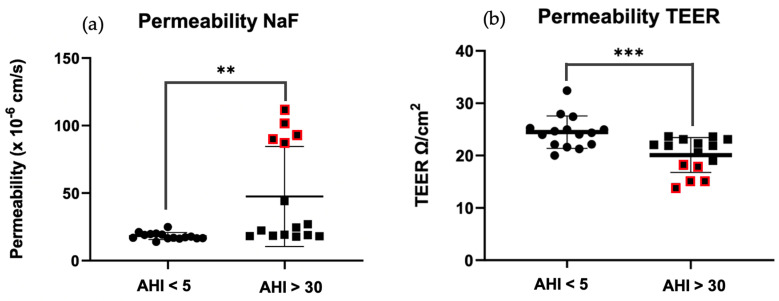
Membrane permeability measurement of HBEC-5i after 5 h of 10 μg exosomes from patients without OSAS AHI < 5 vs. patients with OSAS AHI > 30. (**a**) NaF: sodium fluorescein—results are represented as mean value +/− s.e.m (n = 15 triplicate) ** *p* = 0.0049 < 0.01 between AHI < 5 vs. AHI > 30. (**b**) TEER: transendothelial electrical resistance—results are represented as mean value +/− s.e.m (n = 15 triplicate) *** *p* = 0.0009 < 0.01 between AHI < 5 vs. AHI > 30.

**Figure 5 ijms-25-11058-f005:**
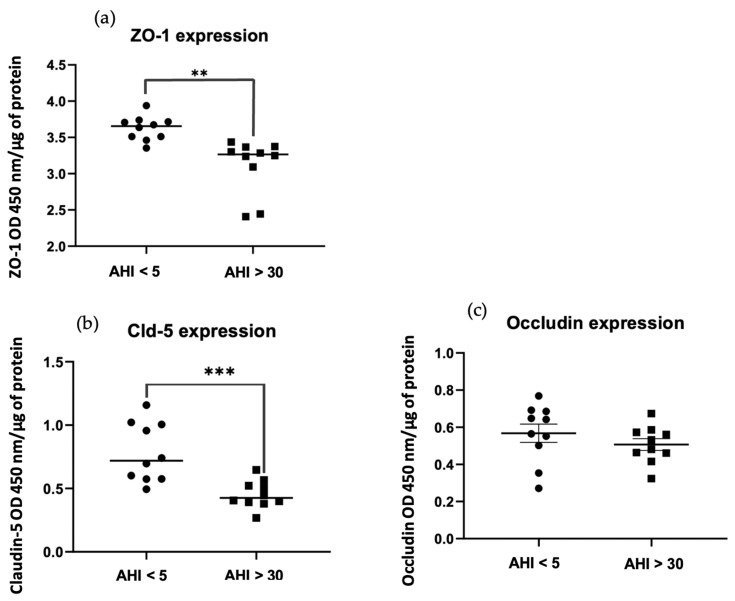
Whole-cell ELISA assay. Results are represented as mean value +/− s.e.m (n = 10 triplicate). Expressions of ZO-1 (**a**) ** *p* = 0.001 < 0.01, claudin-5 (**b**) *** *p* = 0.0008 < 0.01, and occludin (**c**) *p* = 0.3126 > 0.01 proteins after exposure of HBEC-5i 5 h of 10 μg exosomes from patients without OSAS AHI < 5 vs. patients with OSAS AHI > 30.

**Figure 6 ijms-25-11058-f006:**
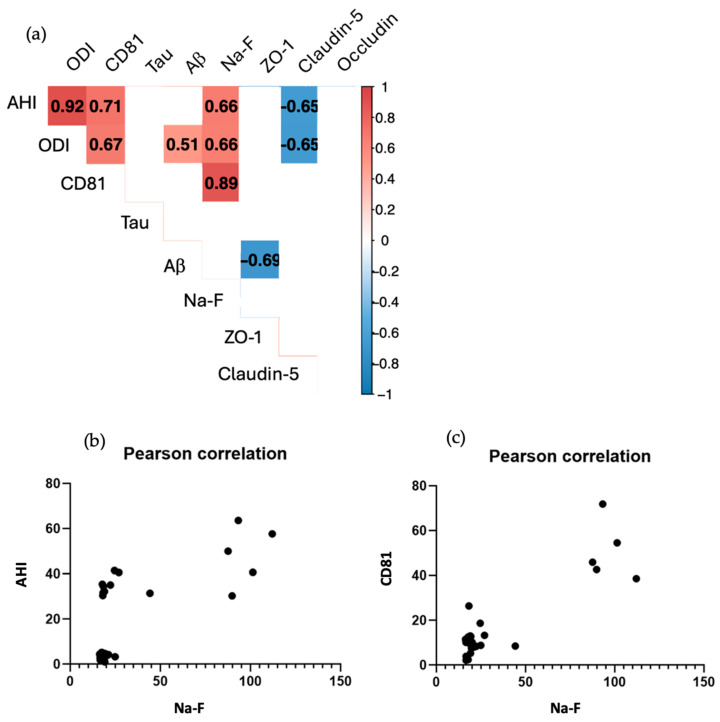
Pearson correlation coefficient R^2^ between parameters: AHI (Apnea Hypopnea Index), ODI (Oxygen Desaturation Index), CD81 tetraspanin, Tau protein, Aβ protein, Na-F permeability, ZO-1 protein, claudin-5 protein, occludin protein. (**a**) Correlation significance tests with a *p*-value < 0.01. Pearson correlation coefficient between Na-F and AHI (**b**). Pearson correlation coefficient between Na-F and CD81 (**c**).

**Table 1 ijms-25-11058-t001:** Descriptive characteristics of the population studies from the PROOF cohort. F: female, M: male, AHI: apnea hypopnea index, ODI: oxyhemoglobin desaturation index, SaO_2_: oxyhemoglobin saturation, LDL: low-density lipoprotein, CRP: C-reactive protein.

Variables	Whole Population	OSAS	No OSAS	*p*
Age (y)	75.8 ± 0.9	75.7 ± 0.9	75.9 ± 0.9	0.621
Sex (F/M)	19/11	9/6	13/2	0.008
AHI (h^−1^)	22.5 ± 21.0	41.2 ± 12.6	3.8 ± 1.5	<0.001
ODI (h^−1^)	14.7 ± 15.1	28.1 ± 9.3	1.4 ± 0.6	<0.001
SaO_2_ min (%)	88.0 ± 5.2	84.3 ± 4.6	91.7 ± 2.1	<0.001
SaO_2_ moy (%)	94.4 ± 1.7	93.4 ± 2.0	95.3 ± 0.7	<0.01
%time SaO_2_ < 90	4.8 ± 11.5	9.6 ± 15.0	0	<0.05
Total cholesterol (g·L^−1^)	2.3 ± 0.4	2.1 ± 0.4	2.4 ± 0.4	0.057
LDL cholesterol (g·L^−1^)	1.4 ± 0.3	1.3 ± 0.3	1.5 ± 0.3	0.119
Glycemia (g·L^−1^)	0.9 ± 0.1	1.0 ± 0.2	0.9 ± 0.1	0.031
CRP (mg·L^−1^)	4.4 ± 7.3	4.6 ± 8.7	4.2 ± 6.0	0.897

**Table 2 ijms-25-11058-t002:** Summary table of the antibodies used.

Antibody	Name	Anti	Reference	Supplier
Primary	CD81 (B-11)	Anti-mouse	SC-166029	Santa Cruz Biotechnology
Primary	CD63 (MX-49.129.5)	Anti-mouse	SC-5275	Santa Cruz Biotechnology
Primary	CD9 (ALB 6)	Anti-mouse	SC-59140	Santa Cruz Biotechnology
Primary	TAU (Tau-13)	Anti-mouse	SC-21796	Santa Cruz Biotechnology
Primary	Amyloid beta (B-4),	Anti-mouse	SC-28365	Santa Cruz Biotechnology
Primary	GAPDH (0411)	Anti-mouse	SC-47724	Santa Cruz Biotechnology
Primary	ZO-1	Anti-rabbit	40-2200	Thermo Fisher
Primary	Claudin-5	Anti-mouse	SC-374221	Santa Cruz Biotechnology
Primary	Occludin	Anti-rabbit	711500	Life tech
Secondary	m-IgGκBP-HRP	Anti-mouse	SC-516102	Santa Cruz Biotechnology
Secondary	Goat anti-Rabbit IgG, HRP conjugate	Anti-rabbit	12-348	Millipore

## Data Availability

Correspondence and requests should be addressed to P.G.

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
