# Peer review of "‘Selected’ Exosomes from Sera of Elderly Severe Obstructive Sleep Apnea Patients and Their Impact on Blood–Brain Barrier Function: A Preliminary Report"

_ijms, 2024, doi:10.3390/ijms252011058_

Round 1
Reviewer 1 Report
Comments and Suggestions for Authors
The study focuses on the impact of exosomes from elderly patients with obstructive sleep apnea syndrome (OSAS) on blood-brain barrier (BBB) function. The article touches on a novel and interesting topic and might bring essential insights into the pathology.
The introduction encompasses the motivation for conducting the study and the hypothesis and the objectives are clearly mentioned. However, I recommend making it more focused and structured as the literature review is too broad.
I recommend specifying exclusion criteria, such as the presence of other neurodegenerative conditions or comorbidities that might influence cognitive function or the blood-brain barrier (BBB). Additionally, there is a gender imbalance between the control and experimental groups, which might affect the results, given that OSA manifests differently in men and women. It would be beneficial for the study to include a detailed explanation of the participant selection process from the PROOF cohort, outlining the exact inclusion and exclusion criteria and the rationale behind them. Also, I suggest clearly stating the study design (e.g., case-control study with experimental in vitro testing) in the methodology, as it is not mentioned.
The conclusions are concise and clear, prompting further research on the subject.
The existing references are somewhat dated. It is imperative to consider implementing revisions and incorporating more recent references, preferably those published within the last five years.
Author Response
Comment 1: (The study focuses on the impact of exosomes from elderly patients with obstructive sleep apnea syndrome (OSAS) on blood-brain barrier (BBB) function. The article touches on a novel and interesting topic and might bring essential insights into the pathology.)
Response 1: Thank you very much for your feedback and for your attention to our article. We have answered your comments point by point and hope you will find it satisfactory. We remain available for any further changes.
Comment 2: (The introduction encompasses the motivation for conducting the study and the hypothesis and the objectives are clearly mentioned. However, I recommend making it more focused and structured as the literature review is too broad.)
Response 2: We agree with your comment and have added a few details to make the introduction more focused (added in orange):
-Presentation of OSA, the risk factors associated with aging and its implication in the development of various disorders including cognitive.
-We defined AHI and ODI and discussed the recommendations that will be useful for understanding our choice of apnea severity later on.
-Presentation of treatment and its effectiveness in limiting cognitive impairment.
-Identifying APNEA patients at high risk of cognitive impairment is difficult.
-One hypothesis is that the BBB may contribute to cognitive problems in patients with OSA.
- Indeed, OSAS influences the permeability of the BBB and could promote the passage of toxic substances to the brain, increasing the risk of neurodegenerative diseases and neuroinflammation (15). This increase in permeability is linked to increased inflammatory signaling and a decrease in tight junction proteins (17)(18).
-A brief presentation of the BBB shows the important components and junctions that enable it to play its role.
-Amyloid beta and tau proteins may be linked to sleep and cognitive problems due to their poor elimination.
IH affects the production and clearance of Ab and tau proteins, linked to neurodegenerative diseases (21) (22). IH increases cytokine production, stimulating beta- and gamma-secretase enzymes, and is responsible for amyloid precursor protein cleavage and Ab accumulation (23) . Furthermore, SF disrupts the cycles required for Ab and tau clearance and alters BBB tight junctions, particularly via Ab peptides (24)(25).-A brief presentation of exosomes.
- Exosomes are found in high quantities in the bloodstream of apneic patients (35) . They represent a valuable source of central nervous system derived biomarkers that can be isolated from blood (36) . The Ab and tau proteins are usually assayed in the CSF. However, the cost of repeating CSF sampling and the invasive nature of this method underlines the importance of developing cheaper and less invasive tests to predict cognitive risks, such as blood tests (27). Ab and tau proteins are very difficult to detect in soluble form in the blood and exosomes could be a good method of assaying these proteins (37).-Explanation of our hypothesis and objectives.
Comment 3: (I recommend specifying exclusion criteria, such as the presence of other neurodegenerative conditions or comorbidities that might influence cognitive function or the blood-brain barrier (BBB). Additionally, there is a gender imbalance between the control and experimental groups, which might affect the results, given that OSA manifests differently in men and women. It would be beneficial for the study to include a detailed explanation of the participant selection process from the PROOF cohort, outlining the exact inclusion and exclusion criteria and the rationale behind them. Also, I suggest clearly stating the study design (e.g., case-control study with experimental in vitro testing) in the methodology, as it is not mentioned.)
Response 3: Of course, we can provide these additional details (added in red and green):
Subjects in this study were selected on the basis of polygraphy findings, taking into account the AHI. We also have biological data such as total cholesterol, LDL, blood glucose and CRP levels. It is important to note that we have no information on whether participants were taking cholesterol- or glucose-lowering drugs. However, assays for these biomarkers were similar between the apneic and control groups, indicating the absence of confounding factors at the time of sampling (Table.1).There was a noticeable imbalance between the number of men and women in the study, as the women were postmenopausal; however, this bias was partly neutralized by the age of the participants. Inclusion criteria were an AHI < 5 for the control group and > 30 for the apneic group, with all subjects in good health, with no signs of inflammation (confirmed by cholesterol, blood glucose and CRP levels). Patients with neurodegenerative diseases or other comorbidities were excluded.
|
LDL cholesterol (g.L-1) |
1.4 ± 0.3 |
1.3 ± 0.3 |
1.5 ± 0.3 |
0.119 |
|
Glycemia (g.L-1) |
0.9 ± 0.1 |
1.0 ± 0.2 |
0.9 ± 0.1 |
0.031 |
|
CRP (mg.L-1) |
4.4 ± 7.3 |
4.6 ± 8.7 |
4.2 ± 6.0 |
0.897 |
We designed a comparative study between two groups of subjects based on the PROOF cohort, in which we had sufficient biological data for 15 patients with severe apnea. To maintain parity between the groups, we randomly selected 15 control subjects from a larger sample. For this first phase of the study, we deliberately chose to compare the two extremes of AHI in order to observe marked differences between the groups. This will facilitate the identification of possible effects of severe apnea.
Further explanation in response to the comment: For this first phase of the study, we deliberately chose to compare the two extremes of AHI in order to observe marked differences between the groups. This will facilitate the identification of possible effects of severe apnea. In a future study, we plan to include patients with moderate and mild apnea to complete our analysis and obtain a more comprehensive view of the impact of different levels of sleep apnea.
Comment 4: (The conclusions are concise and clear, prompting further research on the subject.)
Response 4: Thank you
Comment 5: (The existing references are somewhat dated. It is imperative to consider implementing revisions and incorporating more recent references, preferably those published within the last five years.)
Response 5:
(6): 2022
(14): 2019
(15): 2024
(20): 2024
(21): 2022
(22): 2023
(33): 2018
(35): 2022
(33): 2020
Reviewer 2 Report
Comments and Suggestions for Authors
The study is very interesting, important and novel.
However the major limitation of this study is low number of study participants (only 15 in one group). Therefore Authors have to use the following title "“Selected” exosomes from sera of elderly severe obstructive sleep apnea patients: their impact on the blood-brain-barrier function: a preliminary report". It can generate the interpretation bias. Authors have to describe it as study limitation.
I didn't find information about demographics of the study participants and age and gender comparison between groups. The study included only elderly patients. It can generate the selection bias. Authors have to describe it as study limitation. Authors have to present the demographics of included participants.
The elderly group is specific because they have a lot of comorbidities. Therefore Authors have to present the information about diseases and conditions of study participants and information about medications taken. If the Authors do not have this data, this is another study limitation.
Authors stated that "Our results establish a strong link between sever OSAS and BBB dysfunction in the elderly" based on 15 participants in one group. The strength of statistical analysis performed on such a small group is questionable. Therefore I recommend to write "Our results establish a probable link between sever OSAS and BBB dysfunction in the elderly".
Authors have to provide full description of used statistical tests and provide sample power and size calculation.
Authors have to provide a full description of performed sleep studies. I mean place of sleep assessment, used devices and software as well as international guidelines used for OSA severity assessment.
Authors can consider to present in Introduction the two following important articles related to OSA etiology: doi:10.17219/dmp/185718 and doi:10.17219/dmp/172243
Author Response
Comment 1: The study is very interesting, important and novel.
Response 1: Thank you very much for your feedback and for your attention to our article. We have answered your comments point by point and hope you will find it satisfactory. We remain available for any further changes.
Comment 2: However the major limitation of this study is low number of study participants (only 15 in one group). Therefore, Authors have to use the following title "“Selected” exosomes from sera of elderly severe obstructive sleep apnea patients: their impact on the blood-brain-barrier function: a preliminary report". It can generate the interpretation bias. Authors have to describe it as study limitation.
Response 2: We agree to change the title (added in green):
’Selected‘ exosomes from sera of elderly severe obstructive sleep apnea patients and their impact on blood-brain barrier function: a preliminary report
Comment 3: I didn't find information about demographics of the study participants and age and gender comparison between groups. The study included only elderly patients. It can generate the selection bias. Authors have to describe it as study limitation. Authors have to present the demographics of included participants.
Response 3: Concerning the demographic data of the participants included, we can justify this with the following sentence: “The blood sera used was from the PROOF cohort study in Saint-Etienne (France). This cohort was set up in 2001 and recruited 1011 subjects aged 65 years at the time of inclusion.” The reference doi: 10.1159/000108914 "Autonomic nervous system activity and decline as prognostic indicators of cardiovascular and cerebrovascular events: the 'PROOF' Study. Study design and population sample. Associations with sleep-related breathing disorders: the 'SYNAPSE' Study" explains this in more detail (A prospective cohort of elderly subjects aged 65 years upon study entry were recruited from the electoral list of the city of Saint-Etienne, France. Three initial 2-year examination programs were scheduled for 7 years (2001-2007), followed by late events monitoring.)
The study only included elderly patients. This may generate a selection bias. In fact this is why we have written this sentence in the limitations section of the discussion: “However, our study is limited by the fact that our target population is the elderly. It might be relevant to compare our results with a cohort of younger OSAS patients.”
We hope this answer is sufficient. We can add more details if needed.
Comment 4: The elderly group is specific because they have a lot of comorbidities. Therefore authors have to present the information about diseases and conditions of study participants and information about medications taken. If the Authors do not have this data, this is another study limitation.
Response 4: Indeed, the second reviewer asked us a similar question and we proposed the following additions (added in red and green):
Subjects in this study were selected on the basis of polygraphy findings, taking into account the AHI. We also have biological data such as total cholesterol, LDL, blood glucose and CRP levels. It is important to note that we have no information on whether participants were taking cholesterol- or glucose-lowering drugs. However, assays for these biomarkers were similar between the apneic and control groups, indicating the absence of confounding factors at the time of sampling (Table.1).There was a noticeable imbalance between the number of men and women in the study, as the women were postmenopausal; however, this bias was partly neutralized by the age of the participants. Inclusion criteria were an AHI < 5 for the control group and > 30 for the apneic group, with all subjects in good health, with no signs of inflammation (confirmed by cholesterol, blood glucose and CRP levels). Patients with neurodegenerative diseases or other comorbidities were excluded.
|
LDL cholesterol (g.L-1) |
1.4 ± 0.3 |
1.3 ± 0.3 |
1.5 ± 0.3 |
0.119 |
|
Glycemia (g.L-1) |
0.9 ± 0.1 |
1.0 ± 0.2 |
0.9 ± 0.1 |
0.031 |
|
CRP (mg.L-1) |
4.4 ± 7.3 |
4.6 ± 8.7 |
4.2 ± 6.0 |
0.897 |
We designed a comparative study between two groups of subjects based on the PROOF cohort, in which we had sufficient biological data for 15 patients with severe apnea. To maintain parity between the groups, we randomly selected 15 control subjects from a larger sample. For this first phase of the study, we deliberately chose to compare the two extremes of AHI in order to observe marked differences between the groups. This will facilitate the identification of possible effects of severe apnea.
Comment 5: Authors stated that "Our results establish a strong link between sever OSAS and BBB dysfunction in the elderly" based on 15 participants in one group. The strength of statistical analysis performed on such a small group is questionable. Therefore I recommend to write "Our results establish a probable link between sever OSAS and BBB dysfunction in the elderly".
Response 5: We completely agree with your comment. We have changed our sentence (added in yellow):
Our results establish a probable link between sever OSAS and BBB dysfunction in the elderly.Comment 6: Authors have to provide full description of used statistical tests and provide sample power and size calculation.
Response 6: Thank you for your feedback. We have now added the necessary information (added in pink):
Statistical analysis was realized using unpaired t test by GraphPad software between AHI < 5 and AHI >30 for each of ELISAs, permeability measurements and cell ELISAs. The differences between means were considered to be significant when p values were < 0.05. Results are presented as sample mean +/- SEM.
Statistical analysis was also performed using Pearson correlation coefficient R2 tests between parameters : AHI, ODI, CD81 tetraspanin, tau protein, Ab protein, Na-F permeability, ZO-1 protein, claudin-5 protein, and occludin protein. Only Pearson coefficients with a p-value < 0.01 are displayed and considered significant.
Comment 7: Authors have to provide a full description of performed sleep studies. I mean place of sleep assessment, used devices and software as well as international guidelines used for OSA severity assessment.
Response 7: We agree and propose the following clarifications (added in blue):
Thirty sera were selected for our study: 15 from elderly patients with severe apnea OSAS (AHI > 30) and 15 from elderly patients without apnea and no OSAS (AHI < 5) according to the European Respiratory Society (ERS).
Blood samples were collected in the morning after recording by polygraphy using a polygraphic system (HypnoPTT, Tyco Health-care, Puritan Bennett)(61).
Comment 8: Authors can consider to present in Introduction the two following important articles related to OSA etiology: doi:10.17219/dmp/185718 and doi:10.17219/dmp/172243
Response 8: Very interesting article, but difficult to include in our introduction, especially as the second reviewer asked us to give the introduction more structure. Here is our proposal to include these 2 articles (added in purple and in burgundy):
Added in purple: Indeed, recent research shows that vitamin D deficiency can worsen sleep disorders, particularly in hypertensive patients with OSA, by influencing sleep quality and cognitive function (7).10.17219/dmp/172243
Added in burgundy: Circadian rhythm appears to be closely linked to BBB permeability; and a poor cycle could promote the passage of toxic substances, increasing the risk of neurodegenerative diseases and neuroinflammation (15). In addition, a pilot study on proteins involved in circadian rhythm regulation, revealed that the latter may also play a role in IH-related alterations in glucose metabolism in OSAS patients (16). 10.17219/dmp/185718
Round 2
Reviewer 2 Report
Comments and Suggestions for Authors
The manuscript has been revised correctly. I don't have further comments.